

# Test-retest reliability of a single isometric mid-thigh pull protocol to assess peak force and strength-endurance

Zak Grover[1,*], James McCormack[1,*], Jonathan Cooper[1] and James P. Fisher[1,2]

[1] School of Sport, Health, and Social Sciences, Solent University, Southampton, Hampshire, United Kingdom
[2] Therapeutics, Southampton, United Kingdom
[*] These authors contributed equally to this work.

## ABSTRACT

The purpose of this study was to examine the test-retest reliability of strength-endurance protocols using isometric mid-thigh pull (IMTP). Twenty-eight participants ($23.2 \pm 4.9$ years) completed two protocols across four testing sessions. Protocol one consisted of 10 maximal IMTP tests lasting 5 seconds each with 10 seconds rest between. Protocol two consisted of a prolonged 60 second maximal IMTP. Data from protocol 1 was analysed in two ways; (a) use of the highest peak value from the first three IMTP efforts, and the lowest peak value from the final three IMTP efforts, and (b) use of the mean peak force from the first three IMTP efforts and mean peak force from the final three IMTP efforts. Data from protocol two used the highest and lowest peak values in the first- and final-15 seconds. Analyses revealed excellent reliability for peak force across all four testing sessions (ICC = 0.94), as well as good test-retest reliability for strength-endurance for protocol 1 (a; ICC = 0.81, b; ICC = 0.79). Test-retest reliability for protocol 2 was poor (ICC = 0.305). Bland-Altman bias values were smaller for protocol 1(a = $-8.8$ Nm, b = 21.7 Nm) compared to protocol 2 = (119.3 Nm). Our data suggest that 10 maximal IMTP tests performed as described herein is a reliable method for exercise professionals to assess both peak force and strength-endurance in a single, time-efficient protocol.

## INTRODUCTION

Features of muscular performance include strength (the maximal force capacity of a muscle, or maximal load that can be lifted through a movement, *e.g.*, one-repetition maximum; 1RM), power (ability to apply strength at the highest possible speed), and strength-endurance (ability to resist fatigue by a sustained contraction or perform multiple muscle actions). For the purposes of profiling and evaluation of training adaptation, as well as health assessment, these often form the foundation of testing in athletic (*Burke et al., 2023*; *Lindberg et al., 2022*; *Morrison et al., 2022*; *Pacholek & Zemková, 2020*), non-athletic (*Jonvik et al., 2021*; *Oliveira et al., 2024*), and clinical populations (*Gephine et al., 2021*; *Santagnello et al., 2020*).

There are times when it is beneficial to assess all of these muscular performance measures (*Behm et al., 2021*; *Tamilio et al., 2022*). However, at present these are typically

Corresponding author
James P. Fisher,
james.fisher.phd@outlook.com

done using multiple methods (*i.e.,* separate assessments and/or devices for strength, power, and strength-endurance) through the use of a battery of tests. While this permits specificity for each assessment (*e.g.,* a back squat 1RM, countermovement jump, and maximal repetitions at a submaximal relative load; %1RM), performing multiple tests adds a degree of logistical difficulty—including familiarisation with all protocols, exercise order becomes a factor due to possible fatigue, and testing sessions become more time consuming. It is of course, possible to perform assessment for strength, power, and strength-endurance using a single device (*e.g.,* an isokinetic dynamometer; *Drouin et al., 2004*). Further, while isokinetic dynamometry is typically completed using single-joint movements, multi-joint assessment is possible and is shown to have good test-retest reliability (ICC $=0.75-0.89$; *Callaghan et al., 2000*). However, testing can only be performed unilaterally, there are complexities in machine set-up and learning which make testing large numbers of participants inefficient, and there is often limited accessibility and portability of equipment.

There exists growing popularity in the use of isometric mid-thigh pull (IMTP) devices in assessment of strength and power (rate of force development; RFD) due to correlations between isometric peak force and dynamic peak force at 80, 90 and 100% 1RM ($r = 0.66$, 0.77, and 0.80, respectively, *Haff et al., 1997*), as well as strong relationships with sports performance (*Giles, Lutton & Martin, 2022*; *Ishida et al., 2023*; *Wang et al., 2016*). Furthermore, the high degree of test-retest reliability of the IMTP for peak force (ICC $=0.73-0.99$; *Grgic et al., 2022*) as well as the brevity and simplicity of testing (*Comfort et al., 2019*) and purportedly lower injury risk (*De Witt et al., 2018*) is encouraging for use across a spectrum of populations. However, to date we have found no academic literature considering the use of IMTP in assessing strength-endurance. For the sake of brevity, simplicity and portability, we are investigating as to whether strength-endurance can be reliably tested using IMTP. With this in mind, the aim of this project was to consider the test-retest reliability of different strength-endurance testing protocols using IMTP. The testing protocols have been chosen specifically since data acquired can also be used for assessment of peak force, and power (RFD), though these are not measured in the present study.

## METHODS

### Experimental approach to the problem
All participants attended five sessions, including one familiarization session and four testing sessions (two for each strength-endurance protocol). The familiarization session was used to acquaint each subject with the study protocols and to determine the correct bar height for performance of the IMTP. The four testing sessions were completed in a randomized order separated by 1-week.

### Subjects
Following ethical approval from Solent University (UK) Health, Exercise, and Sport Science (HESS) ethical review board (fishj1HESS2023), 40 male and female recreationally active (*e.g.,* engaging in occasional strength training, sports participation and general activity, but not a member of a sports team or engaging in structured or regular exercise/strength

training sessions) participants were originally recruited based on advertising and snowball sampling through sport and exercise science undergraduate and postgraduate courses. Following a briefing which detailed the research objectives and participant requirements, 37 participants attended a familiarisation session where they completed a PARQ and signed written informed consent which clarified their right to withdraw from the project at any time. All participants met the inclusion criteria which included being aged between 18 and 45, free from musculoskeletal injury or medical conditions which might be aggravated or impact IMTP performance, and self-reportedly free from consumption of anabolic steroids or any other performance enhancing drugs. Exclusion criteria included pregnancy, notable plans to change dietary habits (*e.g.*, Ramadan, *etc.*), or any other cardiorespiratory or medical conditions to which IMTP is contraindicated. Twenty-eight participants (23 men and five women; $23.2 \pm 4.9$ years; 19–42 years; $1.74 \pm 0.11$ m; $76.4 \pm 15.3$ kg, mean $\pm$SD), completed all testing sessions.

## Procedures

During the familiarization session, participants were instructed in the correct position, adopting a hip-width stance, positioning the bar at mid-thigh height, and ensuring an upright torso with a neutral spine. Knee and hip angles were determined within specified ranges ($140 \pm 5°$ and $135 \pm 5°$, respectively) measured using a goniometer (*Moeskops et al., 2018*). During the testing sessions, participants performed a standard, progressive warm-up consisting of 5 min on a cycle ergometer (Monark, Ergomedic 874e) whereby resistance was increased based on personal preference up to ∼60% age-predicted maximal heart rate (*Fisher & Steele, 2017*). Following this each participant completed a practice IMTP where participants were instructed to pull at ∼50% maximal effort, during which time the bar height and hip and knee angles were verified.

Participants attended four testing sessions in a randomised order with 1-week rest in between. Testing was performed at the same time of day and participants were instructed not to change their dietary or caffeine habits on the day of testing, and not to perform any resistance exercise, or strenuous exercise of any kind for 48 h prior to testing days.

Protocol 1 consisted of performing 10 repeated maximal IMTP tests lasting 5 s each, with 10 s rest between. Participants were instructed to pull as hard and as fast as they could, and verbal encouragement to continue exerting maximal effort was provided for each test for each participant (*Engel et al., 2019*), and the 10 s rest interval was timed and reported to the participant with a countdown to begin their next effort. Protocol 2 consisted of a single, and sustained, 60-second maximal IMTP test. Participants were provided the same instructions and verbal encouragement to exert maximal effort, repeated throughout the duration of the test, and time was relayed to the participants every 10 s. Each protocol was performed on two separate occasions. These protocols are similar to those presented by *Duchateau et al. (2002)* who assessed fatigue using 6-second contractions and 4-second rest intervals, compared to a sustained isometric contraction, albeit at 25% MVC in the abductor pollicis brevis muscle. Indeed, different articles have assessed strength-endurance using either prolonged muscle actions (*Waller et al., 2020*), or repetitions of a task (*Vaara*

*et al., 2012*). However, since the IMTP device permits both approaches we chose what was perceived to be two different testing methods.

## Data collection

All tests were performed using VALD dual force plates (ForceDecks, FD4000; VALD Performance, Queensland, Australia) sampling at 1000 Hz, within a VALD isometric mid-thigh pull rack (Queensland, Australia). The force plates were calibrated prior to each trial and then the body mass of each participant measured and subtracted to provide accurate IMTP data, rather than data influenced by body mass which might fluctuate through the duration of the study. Ground reaction force data was recorded from both plates and summed to provide single data value. For protocol 1, peak IMTP values were recorded for all 10 maximal efforts. For protocol 2, the peak value from the first 15 s, and the minimum value from the final 15 s were recorded.

## Statistical analyses

Assessment for normality of distribution was not performed since estimation of intra-class correlation coefficient (ICC) is applicable for uniform and non-uniform data distributions (*Mehta et al., 2018*).

Since protocol one yielded 10 data points for each trial, two practical approaches to analysis were considered to determine maximum and minimum values from which to determine fatigue for strength-endurance testing (referred to as protocols 1a and 1b). The first method included use of the highest peak value from the first three IMTP efforts (max_1a), and the lowest peak value from the final three IMTP efforts (min_1a). The second method involved calculating the mean peak force from the first three IMTP efforts (max_1b) and mean peak force from the final three IMTP efforts (min_1b). Data collection from protocol two automatically yielded highest (max_2) and lowest (min_2) peak values which were used for analyses.

Assessment of test-retest reliability for IMTP was determined for all maximum values (max_1a for trials 1 and 2, and max_2 for trials 1 and 2) using ICC and 95% confidence intervals (CIs).

Strength-endurance was calculated as a fatigue value from each protocol based on maximum minus minimum values for protocols 1a, 1b, and 2, and from this, reliability of strength-endurance was assessed between trials 1 and 2 for each protocol using ICC and 95% CIs. Furthermore, Bland-Altman plots were performed to consider the agreement in fatigue between trials 1 and 2 for each protocol (*e.g.*, 1a, 1b, and 2). Finally, assessment of the reliability of 10 maximal effort IMTP tests (as per protocol 1) across both trials was considered using ICC and 95% CIs.

ICC values <0.5, 0.5–0.75, 0.75 to 0.9, and >0.90 were considered poor, moderate, good, and excellent reliability, respectively (*Koo & Li, 2016*). Analyses were completed using the statistical package for social sciences (SPSS) v29. Bland-Altman plots were performed using BA-plotteR (https://huygens.science.uva.nl/BA-plotteR/).

## RESULTS

Test-retest reliability for IMTP for all maximum values across all trials and protocols (max_1a for trials 1 and 2, and max_2 for trials 1 and 2) was excellent (ICC = 0.94; 95% CI [0.89−0.97]). As additional, exploratory analysis, we performed reliability testing for maximal IMTP data for three of four testing time points, excluding the first testing session attended by each participant. Reliability was excellent (ICC = 0.96; 95% CI [0.93−0.98]).

Test-retest reliability for fatigue for protocol 1a and 1b between trials 1 and 2 was good (1a; ICC = 0.81, 95% CI [0.62−0.91], 1b; ICC = 0.79, 95% CI [0.60−0.90]). Test-retest reliability for fatigue for protocol 2 between trials 1 and 2 was poor (ICC = 0.31; 95% CI [−0.07−0.60], $p > 0.05$). Maximum and minimum values (N), absolute fatigue (N), relative fatigue (minimum/maximum*100; %), coefficient of variation (CV), and reliability values are presented in Table 1.

Bland-Altman plots are shown as Figs. 1−3, revealing bias and 95% limits of agreement (LoA) of protocol 1a = −8.8N, LoA = −279.8 to 262.3N, protocol 1b = 21.7N, LoA = −193.4 to 236.8N, and protocol 2 = 119.3N, LoA = −497.2 to 735.8N). Notably the $y$-axis for Fig. 2 (protocol 1b) crosses 0 on the $x$-axis, this is a result of some negative values for fatigue data from protocol 1b (*i.e.,* the mean peak force from the first three IMTP efforts (max_1b) was lower than the mean peak force from the final three IMTP efforts (min_1b)).

A final assessment to investigate the reliability of each participant across all 10 maximal effort IMTP tests (as per protocol 1) across both trials, revealed excellent reliability (ICC = 0.97; CIs = 0.96 to 0.97).

## DISCUSSION

The aim of this study was to evaluate the test-retest reliability of strength-endurance testing protocols using IMTP. Our data produced some notable findings that suggest the use of IMTP testing can be reliable for strength and strength-endurance assessment in a single testing session. Primarily, we should recognise the high test-retest reliability of the IMTP. Maximal values for participants across four different testing trials showed excellent reliability (ICC = 0.94; 95% CI = 0.89−0.97), comparable to that presented in a previous review (ICC = 0.73−0.99; *Grgic et al., 2022*). This high test-retest reliability suggests that there is minimal learning effect from practicing IMTP tests. However, in an effort to accurately assess this we performed further exploratory analysis of our data by removing the first maximal testing session for each participant. The between-day reliability of IMTP then increased to ICC =0.96 (95% CI = 0.93−0.98), suggesting that any learning effect from the first testing session is small.

In consideration of the protocols to assess test-retest reliability of strength-endurance, data from protocol 1 (10 maximal IMTP efforts interspersed with a short rest interval; ICC = 0.79−0.81) showed greater reliability compared to protocol 2 (a prolonged 60-second maximal effort; ICC = 0.31). The degrees of fatigue also differed considerably between protocols (protocol 1 = 7.7–15.3%, and protocol 2 = 28.3–33.3%), which are to be expected since one would assume that a prolonged maximal contraction (*i.e.,* protocol 2) would induce greater fatigue compared to repeated maximal efforts with rest intervals (protocol

 

Grover et al. (2024), *PeerJ*, DOI 10.7717/peerj.17951

**Table 1  Mean (±SD) values (N) for Maximum, Minimum, and Fatigue values as well as relative fatigue (% of max) and coefficient of variation (CV) from trials 1 and 2 for each protocol.**

| Protocol | Trial 1 | | | | | Trial 2 | | | | | ICC (95% CIs) |
| --- | --- | --- | --- | --- | --- | --- | --- | --- | --- | --- | --- |
| | Max peak | Min peak | Fatigue | Fatigue (%) | CV (%) | Max peak | Min peak | Fatigue | Fatigue (%) | CV (%) | |
| 1a | 2001.8 ± 568.3 | 1699.9 ± 455.2 | 302.0 ± 232.5 | 15.1 | 1.3 | 2029.9 ± 568.3 | 1719.2 ± 449.3 | 310.7 ± 211.9 | 15.3 | 1.5 | 0.81; 0.62–0.91 |
| 1b | 1925.2 ± 549.1 | 1752.8 ± 474.5 | 172.4 ± 177.6 | 9.0 | 1.0 | 1945.7 ± 533.5 | 1795.0 ± 482.2 | 150.6 ± 163.0 | 7.7 | 0.9 | 0.79; 0.60–0.90 |
| 2 | 1931.6 ± 542.9 | 1287.5 ± 378.1 | 644.1 ± 265.6 | 33.3 | 2.4 | 1853.1 ± 504.2 | 1328.3 ± 376.5 | 524.8 ± 267.8 | 28.3 | 2.0 | 0.31; −0.07–0.60 |

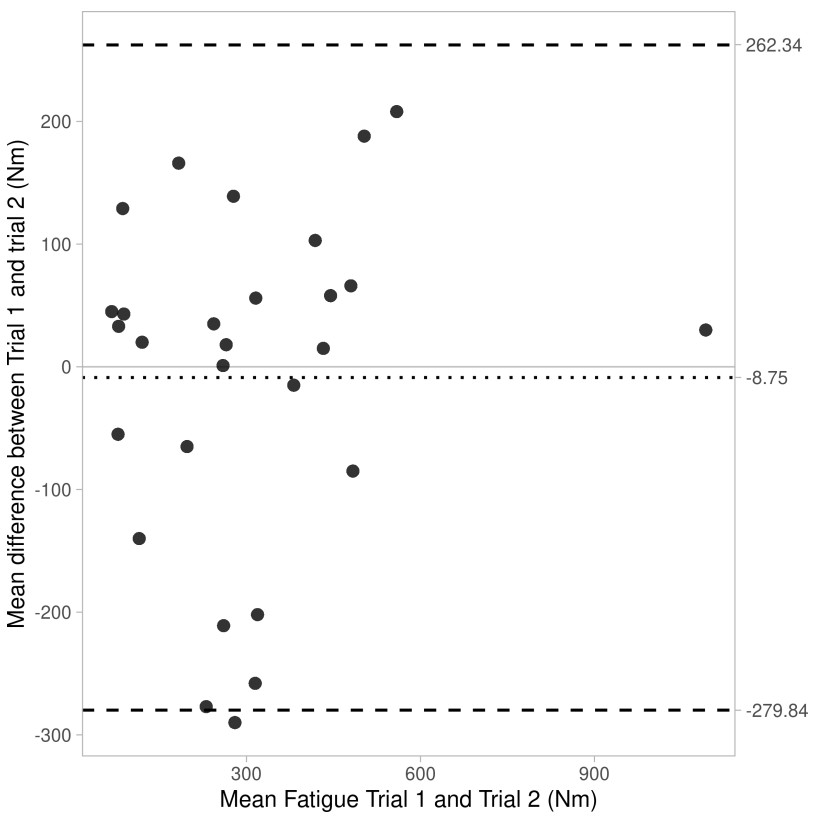

**Figure 1** Bland Altman plot showing agreement (bias and 95% CIs) for fatigue between trials 1 and 2 for Protocol 1a.

1). However, this data is presented descriptively and without analyses for significance. It is worth acknowledging that strength-endurance can be affected by psychological factors (*e.g.*, boredom and motivation), and that this might be greater in a single prolonged muscle contraction compared to shorter, repeated maximal strength tests (*Micheletti et al., 2021*). This is likely true even despite strength-endurance test-retest reliability being greater with verbal encouragement (*Engel et al., 2019*). Certainly, the higher reliability values for protocol 1 are more consistent with the body of literature considering strength-endurance test-retest reliability for maximal number of repetitions possible at a submaximal load (ICC $=0.98-0.99$; 17 and ICC $=0.87-0.95$; *Mann et al., 2014*). The poor reliability of protocol 2 was supported by Bland-Altman analyses revealing poor agreement in fatigue values between trials 1 and 2, as well as large limits of agreement (see Figs. 1–3).

In considering the use of isokinetic dynamometry for strength-endurance testing, reliability appears to vary and might be speed dependent. For example, *Pincivero, Lephart & Karunakara (1997)* reported poor reliability (ICC = $0.52-0.74$) for performance of 30 repetitions at 180°/sec, and *Montgomery, Douglass & Deuster (1989)* reported similarly low ICC values (<0.80) at the same speed. In contrast, *Manou et al. (2002)* reported better reliability ($r > 0.81$) for strength-endurance testing following 40 maximal knee extension and flexion effort at 120°/sec. Notably, the research using isokinetic devices for
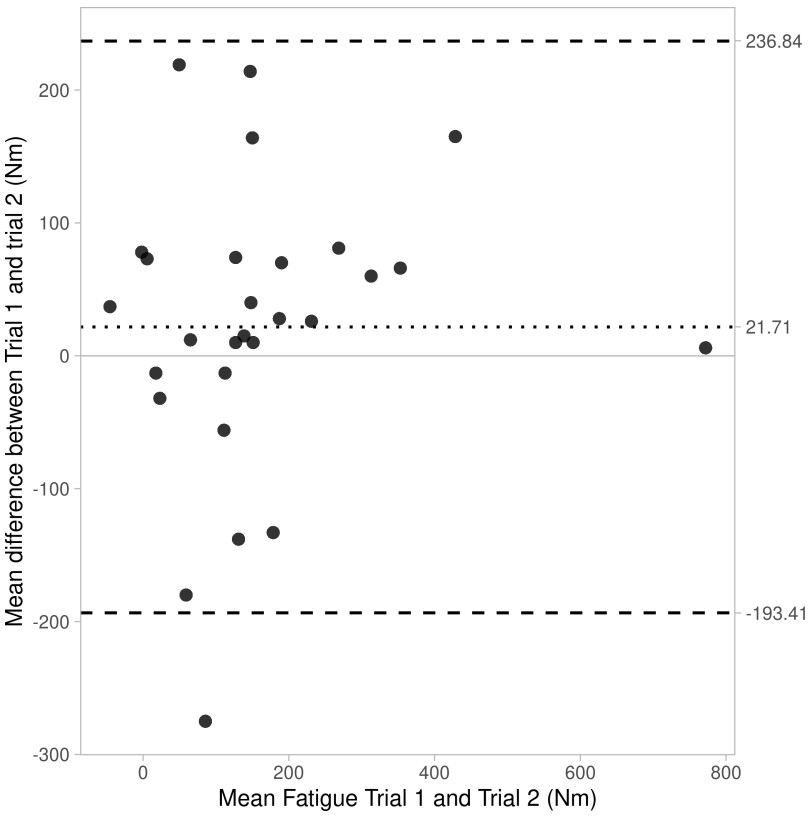

**Figure 2    Bland Altman plot showing agreement (bias and 95% CIs) for fatigue between trials 1 and 2 for Protocol 1b.**

strength-endurance testing is all unilateral and single-joint in nature. The data presented for protocol 1 for IMTP strength-endurance testing herein (ICC = 0.79−0.81) seems comparable to the highest values reported for isokinetic dynamometry but might present greater ecological validity being multi-joint and bilateral, as well as being more practical based on convenience of set-up and portability of equipment.

While our analyses has established that repeated maximal contractions produce more reliable fatigue compared to a single prolonged contraction, we should also consider the different data analyses performed for protocol 1 (*i.e.,* 1a and 1b). Data for protocol 1a might initially appear limited; by taking the highest peak value from the first three IMTP efforts (max_1a), and the lowest peak value from the final three IMTP efforts (min_1a), there is a risk that any single value is a combination of observation and random error (*Bland & Altman, 1996*). Thus, we also chose to analyse the mean peak force from the first three IMTP efforts (max_1b) and mean peak force from the final three IMTP efforts (min_1b) as an alternative to calculate fatigue. However, our test-retest reliability of maximal values was excellent (ICC = 0.94) suggesting maximal effort from participants in all trials, and minimal error in these observation values. In turn, the use of mean peak force from the first and final three IMTP tests is evidently limited by mean peak values being higher in the final three compared to the first three IMTP tests for some participants.[1] We could

[1] For trial 1, this occurred in three of 28 participants with values of −27, −52, and −31N, and for trial 2 this occurred for four of 28 participants with values of −64, −41, −60, and −31N. This observation was only consistent across both trials for one participant, and we consider these low compared to the SD values of 177.6N and 163.0N for trials 1 and 2.

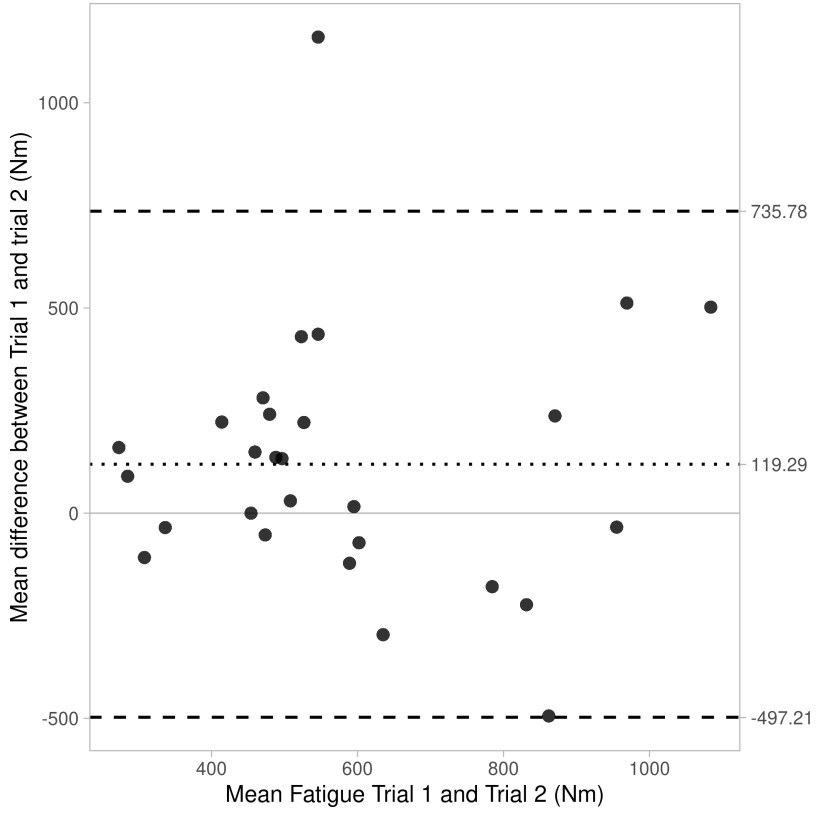

**Figure 3** Bland Altman plot showing agreement (bias and 95% CIs) for fatigue between trials 1 and 2 for Protocol 2.

surmise that this is suggestive of participants not providing maximal effort throughout all 10 tests and thus not incurring the same fatigue and decrement in force for the mean of the final three IMTP tests. Further, there might be truth of a "sprint finish" effect whereby knowing they were nearing completion of the 10 maximal tests they were able to apply additional effort. However, these are both purely supposition and contradict guidance to participants to provide maximal effort for every IMTP test. In addition, and in an effort to assess whether any nonconformity to providing maximal effort might have occurred, we completed additional test-retest analysis for all 10 IMTP tests for all 28 participants (280 data points for each trial), revealing excellent test-retest reliability (ICC = 0.97; CIs = 0.96 to 0.97).

Finally, we should also recognise potential limitations of the current project. This is the first research study to consider assessment of strength-endurance using IMTP, and while we would not expect the results to differ for other population groups, we cannot assume the same reliability. We suggest future research should also consider the assessment of all muscular function parameters in a single protocol. For example, we present a practical approach to assessing peak force and strength-endurance through the use of 10 maximal IMTP tests, however, in the same protocol a tester can also measure power through RFD,[2]

[2] Although recent data suggests that RFD reliability can vary based upon time; for example, a recent study by *Ishida et al. (2023)* reported ICC (and 95% CIs) of 0.76 (0.52−0.89), 0.94 (0.86−0.97), and 0.95 (0.90−0.98) for RFD at 90, 200, and 250 ms, respectively.

and also power-endurance by assessing any decline in RFD through the 10 tests in a similar format to that presented herein.

## Practical applications

Sport scientists, strength coaches, and exercise professionals often perform maximal tests of muscular performance with a range of athletic, non-athletic, and clinical populations. However, the requirement to complete a variety of tests is not always appropriate. We present a single protocol for IMTP, with good test-retest reliability, which can assess peak force and strength-endurance and provide data for muscular power. We invite exercise science professionals to use this practical and time efficient method to assess whole body muscular performance in the future.

## ACKNOWLEDGEMENTS

The authors wish to thank the participants for their involvement in this research project.

### Funding

The authors received no funding for this work.

### Competing Interests

The authors declare there are no competing interests.

### Author Contributions

- Zak Grover conceived and designed the experiments, performed the experiments, analyzed the data, prepared figures and/or tables, and approved the final draft.
- James McCormack conceived and designed the experiments, performed the experiments, analyzed the data, prepared figures and/or tables, and approved the final draft.
- Jonathan Cooper conceived and designed the experiments, prepared figures and/or tables, authored or reviewed drafts of the article, and approved the final draft.
- James P. Fisher conceived and designed the experiments, performed the experiments, analyzed the data, prepared figures and/or tables, authored or reviewed drafts of the article, and approved the final draft.

### Human Ethics

The following information was supplied relating to ethical approvals (i.e., approving body and any reference numbers):

Solent University, Southampton Health, Exercise and Sport Science ethics committee granted Ethical approval to carry out the study within its facilities (Ethical Application Ref: fishj1HESS2023).

### Data Availability

The raw measurements are available in the Supplementary File.

## Supplemental Information

Supplemental information for this article can be found online at http://dx.doi.org/10.7717/peerj.17951#supplemental-information.

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
