# Peer review of "Test-retest reliability of a single isometric mid-thigh pull protocol to assess peak force and strength-endurance"

_PeerJ, doi:10.7717/peerj.17951_

## Round 0.1 · original submission · Major Revisions

Our two expert reviewers have identified several areas for improvement in your manuscript - with both querying aspects of experimental design. Please take time to provide a clear response to all comments; we look forward to receiving your revision.

·

Basic reporting

The authors have conducted an interesting study and I liked the approach of coming up with an assessment that could potentially be used to assess multiple aspects of strength and strength endurance. The manuscript is well written and easy to follow. It would be good if the authors could cite some of the original research that was performed using the isometric mid thigh pull. For example, see the work done by Dr Mike Stone and Dr Greg Haff such as Haff et al. (1997) Force-time dependent characteristics of dynamic and isometric muscle actions. JSCR 11(4): 269-272.

Experimental design

Some aspects of the experimental design do require some clarification and further justification.
The choice of protocols were interesting and seem appropriate for the measurement of strength endurance. I did wonder about the approach taken for the assessment of peak force. Including an additional testing session where the more traditional method of peak force assessment was used would have strengthened the study in my opinion e.g. several maximal attempts with longer rest periods such as 2-3 attempts with 3-5 minutes rest. The authors should include some further justification for protocol 1 where they used 10 x 5 second efforts for the IMTP. This appears to be based off the work of Duchateau et al. that used a small muscle group, and it is not clear how this translates to the IMTP.

Validity of the findings

The work is clearly novel and adds the current of literature on the use of the IMTP. Practical and robust approaches have been undertaken for determination of the reliability. However, I believe it would be useful to also include CV’s as an additional measure of reliability given that these are commonly reported for tests such as the IMTP. This should be a relatively straightforward analysis to perform and will complement the other measures that have been presented.

Additional comments

Title Its not a major point but I wonder if “strength endurance” rather than “muscular endurance” would be a more appropriate term for the title and throughout the manuscript?
Line 80 Specify here that it was peak force that was analysed in the review by Grgic and colleagues.
Line 96 Provide more detail on the background of the participants. It is not clear what is meant by “recreationally active” and it is not mentioned in the inclusion criteria.
Line 134 Include the specific type of isometric contraction that was used in the Duchateau study i.e. abductor pollicis brevis.
Line 140 Further explanation is needed here on why subtracting body mass provides more “accurate” IMTP data.
Line 178-9 Change Nm to N.
Line 240 I’m not sure that “deviance” is the best word here?
Line 251 What about the reliability of RFD? Previous research suggests that this is questionable (e.g. Ishida et al. 2023. JSCR 37: 18-26) so I would recommend reconsidering this part of the discussion “….however, in the same protocol a tester…”
Table 1 and Figures Change Nm to N.

·

Basic reporting

Reporting and standard of scientific writing is straight forward and clear. Would say as for below comments that the background and context could be better explained.
Introduction; I feel the reasoning for the study is lacking somewhat- Especially the reasoning as to why one IMTP was tested against another compared to another method, such as dynamometry which is mentioned? Is it possible to please allude to this and clarify the reasoning somewhat in the introduction section to highlight the implication and application of this work better? Specifically, the study introduction does not really lead into the reasoning for assessing muscular endurance via IMTP? Why might this provide a better or similar option beyond logistical reasons is not fully explained I feel.
Line 68- Learning effect is alluded to here which is indeed an issue across all these types of measures with neuromuscular function being rather “elastic” so to speak. What is the learning effect data for IMTP? There is literature examining the learning effects of various neuromuscular function assessments, for example Rittweger et al. (2004); https://doi.org/10.1111/j.1532-5415.2004.52022.x is it perhaps instructive to also include this literature either here, or perhaps more appropriately in the discussion, to allow the reader to fully understand the implications and limitations of the assessments that are proposed here? You also mention later in the introduction that test-retest is high for the IMTP, but would suggest it be further added to here or in the discussion, as I believe this strengthens the argument for this measure, amongst a lot of other measures which have a significant learning effect.

Experimental design

Methods are nicely described and very simple throughout, however can “recreationally active” be better defined? The data shows about a 25% variation in the participants peak force output, which is quite large, suggesting some participants were more highly trained than others? Could this possibly impact the outcomes in terms of fatigue assessment? Also those who are highly resistance becoming more fatigued depending on the initial peak force compared to those primarily taking part in endurance activities?
Also in the method, participant warm up was “up to” 60% Age Related maximal HR. Please could I ask this to be defined a little better, were participants stopped at 60%? Or carry on for the full 5mins? In terms of 50% maximal effort on the practice also, was this based on RPE? Could do with more definition here I feel.

Validity of the findings

The discussion is well rounded and hypothesises with evidence, but feel if the rationale for the study is using this method over dynamometry or other muscular fatigue assessments, these need to also be compared here? Ie, how does this compare to the test-retest of multiple fatiguing contractions using dynamometry methods? Appreciate this is single joint, but you also mention this in the introduction, so would be a worthwhile comparison for the reader between these options I would argue. Please discuss, as you discuss your findings well, but only your findings I would argue and to truly compare your data to others, the ICC’s should be compared to those of other methods if you are advocating the use of IMTP as a valid and applicable measure?
As above comments in terms of impact and novelty assessment, rationale and specific applications I feel could be better explained and evidence based.

---

## Round 0.2 · accepted · Accept

The changes made have significantly improved the manuscript, addressing all points from the reviewers. The manuscript is now ready for publication.

·

Basic reporting

No comment

Experimental design

No comment

Validity of the findings

No comment

Additional comments

Thank you to the authors for completing the study and making the suggested changes.